# Social familiarity improves fast-start escape performance in schooling fish

Lauren E. Nadler [1,2,5]✉, Mark I. McCormick[1], Jacob L. Johansen[3] & Paolo Domenici[4]

Using social groups (i.e. schools) of the tropical damselfish *Chromis viridis*, we test how familiarity through repeated social interactions influences fast-start responses, the primary defensive behaviour in a range of taxa, including fish, sharks, and larval amphibians. We focus on reactivity through response latency and kinematic performance (i.e. agility and propulsion) following a simulated predator attack, while distinguishing between first and subsequent responders (direct response to stimulation versus response triggered by integrated direct and social stimulation, respectively). In familiar schools, first and subsequent responders exhibit shorter latency than unfamiliar individuals, demonstrating that familiarity increases reactivity to direct and, potentially, social stimulation. Further, familiarity modulates kinematic performance in subsequent responders, demonstrated by increased agility and propulsion. These findings demonstrate that the benefits of social recognition and memory may enhance individual fitness through greater survival of predator attacks.

[1] ARC Centre of Excellence for Coral Reef Studies, James Cook University, Townsville, QLD, Australia. [2] College of Science and Engineering, James Cook University, Townsville, QLD, Australia. [3] Hawai'i Institute of Marine Biology, University of Hawai'i at Manoa, Kaneohe, HI, USA. [4] CNR-IBF, Institute of Biophysics, Pisa, Italy. [5]Present address: Department of Marine and Environmental Sciences, Nova Southeastern University, Dania Beach, FL, USA. ✉email: lnadler@nova.edu

In stable social groups, familiarity develops through repeated interactions among individuals[1], allowing them to acquire knowledge of group-mates' behaviour in various contexts (e.g. feeding, defence) and to develop an individualised role within their group[2,3]. As such, familiarity maximises many of the benefits of sociality, by minimising aggression, enhancing cooperative foraging, improving growth rates, providing mating opportunities and increasing reproductive output[4–7]. Unfamiliar individuals, conversely, pose a number of threats to a stable group, including exposure to novel pathogens, harm to offspring, or competition for resources[8,9]. Empirical evidence from a range of gregarious species (e.g. primates, cetaceans, rodents) indicates that individual fitness can be enhanced by familiarity, with longer lifespans and stronger immune function observed[10]. Prior social experience also enhances trust in conspecific alarm cues. For instance, in diverse species including macaques and squirrels, individuals are more responsive to the auditory alarm vocalisations of familiar individuals than unfamiliar conspecifics[11,12]. These effects are likely to improve anti-predator behaviours. Spider mites living in a familiar social structure, for example, react more quickly to predator attacks and survive a greater number of predator encounters than unfamiliar social groups[13]. Hence, association preferences are frequently shaped by familiarity[7,14–17], overriding the tendency to group with conspecifics and kin[18–20].

In fish groups (i.e. schools), individuals depend on social cues to survive. One of the main forms of defence from predation in fish is the fast-start response, which is a rapid, anaerobically driven acceleration, typically in response to a threat stimulus[21]. Following a predator attack, individuals in social groups can be alerted to predation risk either by directly detecting the predator's presence or through socially transmitted information from group-mates, typically in the form of rapid changes in locomotion (e.g. speed and direction)[22–25]. The fast-start response is commonly initiated by a pair of higher-order command neurons called Mauthner cells (M-cells)[26–29]. If M-cell functionality is lost through experimental ablation, rapid fast-start responses (e.g. latency <16 ms from the stimulus onset, as measured in ref. [22]) are absent, though longer latency, slower responses can be initiated by other reticulospinal neurons in the brainstem escape network[29]. However, there are direct links among survival of predator attacks, reaction timing and escape kinematics, with the chance of survival increasing directly with faster reaction time, higher speed, and greater acceleration[29–31]. Despite the abundant evidence for the benefits of familiarity to processes including defence[32,33], there remains a gap in our understanding about the mechanisms driving these effects, and, particularly, whether they stem from changes in individual behaviour, social communication, or some combination of individual and social factors.

All animals aim to balance the risk of predation against the energy investment necessary to execute an escape, to maximise the number of correct reactions (e.g. reacting to the presence of a predator) and minimise reactions to inaccurate information (e.g. reacting to harmless stimuli)[34–36]. From the social-communication perspective, schooling fish may modulate their sensitivity to social information with the level of familiarity among the individuals in the group, due to greater certainty in information accuracy once familiarity is achieved[36]. Familiarity may also promote a more effective school structure (i.e. greater cohesion)[32], boosting collective sensitivity to risk in social groups[33,37] and enhancing the interaction network among neighbours[25].

On an individual level, an unfamiliar social context may have behavioural and/or physiological impacts that either delay or prevent initiation of the M-cell, with implications for reaction timing and escape kinematics. For example, conspecific inspection in unfamiliar groups could tax the cognitive process of awareness, limiting the attention available for vigilance and delaying M-cell initiation[38]. Further, if the presence of unfamiliar conspecifics generates a physiological stress response (e.g. spikes in cortisol and other glucocorticoid stress hormones)[39,40], neuronal inhibition can be a consequence[41,42], in which the stimulus threshold necessary for M-cell firing could be modified, increasing the level of threat necessary to initiate a rapid fast-start response.

Here, we tested how familiarity influences the reactivity, agility and propulsive performance of fast-start escape responses of "first responders" (i.e. direct response to stimulation) and "subsequent responders" (i.e. followers whose response could be triggered by direct and/or social stimulation), using schools of the tropical damselfish Chromis viridis (Pomacentridae). In addition, we measured how school cohesion and coordination changes during the first 100 ms following a simulated predator attack, a crucial time period for avoiding predation[30,31]. Chromis viridis are obligate live coral dwellers that live in schools ranging in size from three to hundreds of individuals[43–45]. Like other schooling fishes, C. viridis are likely to encounter unfamiliar individuals through fission-fusion events. Individuals may choose to migrate to nearby groups that: (1) exhibit phenotypes better matched to their own, (2) host spawning aggregations or (3) live on a habitat with preferred characteristics[46]. Data on C. viridis at the study site indicate that the majority of individuals maintain fidelity to a single school, while a minority migrates to alternative groups up to 80 m away (Nadler, Killen, Cox and McCormick, unpublished data). In addition, coral reefs are subject to a range of disturbances, such as storms and habitat fragmentation, which can lead to group disruption and either forced association with unfamiliar schools or social isolation[47,48]. We hypothesised that familiarity would aid in the individual performance of both first and subsequent responders as well as whole-school performance following a simulated predator attack, as repeated and frequent interactions among individuals incentivize cooperation, reciprocity, and exchange of resources among group-mates[49].

## Results

**Individual fast-start escape performance.** Schools composed of eight familiar or unfamiliar C. viridis individuals ($n = 12$ schools per treatment, referred to as familiar and unfamiliar schools, respectively, hereafter) were tested in a laminar flow tank, using a current speed (3.2 cm/s) that mimicked their natural habitat conditions on a calm weather day[50]. Prior to testing, all experimental schools were assembled from equally unfamiliar fish, by joining eight individuals from eight geographically distinct wild schools (schools separated by a minimum of 100 m). Prior to escape response testing, we confirmed that our study species achieves familiarity in a comparable time frame to another tropical fish species, the guppy Poecilia reticulata, using a choice test methodology[51] (Supplementary Fig. S1). Schools from the "familiar" treatment were given a period of 3 weeks to familiarise prior to testing. Unfamiliar schools were assembled immediately prior to testing.

The arena inside the laminar flow tank allowed fish (standard length: $3.33 \pm 0.02$ cm, mean ± s.e.) to complete a full escape response without interference from the walls of the arena (50 cm L × 40 cm W × 9 cm D)[52]. Escape responses were elicited using a mechanical stimulus simulating an aerial predator attack[53], released using an electromagnetic switch once ≥75% of the individuals in the group were >1 standard body length (L) from the swim tunnel walls and within four L of the stimulus. Responses were video recorded using a high-speed camera (240 fps).

We quantified fast-start characteristics associated with the initial unilateral body bend following stimulation (i.e. stage 1 of

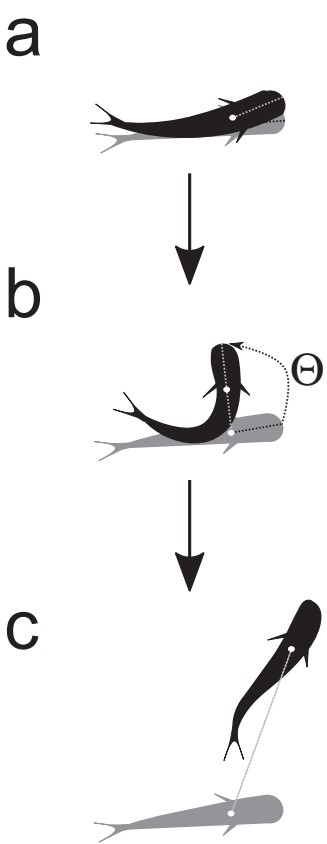

**Fig. 1 Components of the fast-start escape response.** This study examined the role of social familiarity in the reaction timing and kinematic performance of schools of the tropical damselfish *Chromis viridis*, focusing on three key traits: latency, average turning rate, and distance covered. **a** Latency indicates the time period between the threatening stimulus breaking the water surface and the fish's first movement, with a shorter latency indicating a faster reaction time. **b** Average turning rate is measured by dividing the angle achieved during the first unilateral bend of the reaction (i.e. stage 1) by the duration of time to achieve that angle, with a higher turning rate indicating greater agility in the response. **c** Distance covered indicates the distance moved during the first 42 ms of the reaction, the mean time period for individuals to complete two body bends (i.e. stages 1 and 2), and is indicative of the response's speed and acceleration. In all panels **a**–**c**, the grey fish silhouette indicates the fish's position immediately prior to stimulation and the black fish indicates the fish's position during each component of the fast-start escape response.

the fast-start) and the subsequent contralateral body bend (i.e. stage 2 of the fast-start)[21]. Individual fast-start escape performance was characterised through traits associated with reaction timing and kinematic performance, including latency (the time period between the threatening stimulus making contact with the water surface and the fish's first movement, Fig. 1a), average turning rate (the ratio of the turning angle achieved during stage 1 to the stage 1 duration, Fig. 1b) and distance covered (distance moved by the centre of mass during the first 42 ms of the reaction, the mean time to achieve stages 1 and 2 in this species according to published data[54], Fig. 1c). Individuals were assigned a responder number based on the sequential order of latencies in their respective groups. Responder number was designated using standard competition ranking, such that if two individuals tied for first place, the responder numbers for that group would be 1, 1, 3, 4, 5, 6, 7, 8 (sample sizes for each responder number by treatment are detailed in Table S1; ties occurred in 13 out of

24 schools tested; number of ties: 2–4 individuals per school; number of schools exhibiting ties by treatment: $n = 9$ familiar schools, $n = 4$ unfamiliar schools). While the responses for non-first responders (i.e. responder numbers 2 through 8) could have been initiated either directly by the stimulus or indirectly by their group-mates' responses, first responders' responses could only have been initiated directly by the stimulus (i.e. this fast-start behaviour was never observed outside of the post-stimulus time period).

Analyses indicated that all aspects of individual escape performance improved significantly with familiarity, with effects found for some traits in subsequent responders, and some traits in both first and subsequent responders (complete model output detailed in Supplementary Tables S2–S4). Specifically, we found that the first responders in familiar groups exhibit latencies that are approximately 60% shorter than first responders in unfamiliar groups (Generalised Linear Model (GLM): $F_{1,22} = 9.2$, $p = 0.007$, $R^2 = 0.43$; point biserial correlation coefficient of first responder latency vs. school type $= -0.24$; Fig. 2a) and are hence reacting more quickly to the predator stimulus (familiar: 7.5 ± 0.9 ms, unfamiliar: 12.5 ± 1.5 ms, mean ± s.e.). This effect extended to subsequent responders (responder numbers 2–8) as well, with subsequent responders from familiar schools (i.e. familiar fish) exhibiting a nearly five times faster latency overall when compared to those from unfamiliar schools (i.e. unfamiliar fish) (familiar: 67.7 ± 17.7 ms, unfamiliar: 308.0 ± 39.9 ms, mean ± s.e.; Linear Mixed Effects Model (LMM): $F_{1,22} = 12.3$, $p = 0.002$; marginal $R^2$, $R^2m = 0.51$, conditional $R^2$, $R^2c = 0.86$; Fig. 2b). Further, there was a significantly greater proportion of familiar than unfamiliar fish (46% vs 25%) that responded to stimulation with a latency <16 ms, i.e. a time lag consistent with latencies in M-cell initiated responses in other species[22] (LMM: $\chi^2 = 8.0$, $p = 0.005$, $R^2m = 0.14$, $R^2m = 0.33$).

Although there was no difference in average turning rate with familiarity in first responders (Supplementary Table S3A), average turning rate was more than 50% higher with familiarity in subsequent responders, when compared to those from unfamiliar groups (LMM: $F_{1,22} = 7.3$, $p = 0.01$, $R^2m = 0.29$, $R^2m = 0.58$; Fig. 3; familiar: 5530 ± 247°/s, unfamiliar: 3480 ± 289°/s, mean ± s.e.). This result indicates that familiar subsequent responders exhibit greater agility while escaping that would steer the animal away from the striking predator's trajectory[55]. Subsequent responders across both familiarity treatments declined in their average turning rate with rising responder number ($F_{1,131} = 21.6$, $p < 0.0001$), indicating that the strongest average turning rate in both familiarity treatments occurred in the earliest responders following the stimulus, with this performance metric dropping off as responder number increased. Stimulus distance also had a marginal effect on average turning rate, declining as stimulus distance increased across both familiarity treatments ($F_{1,142} = 3.35$, $p = 0.07$; Supplementary Fig. S3A).

As with the average turning rate, the distance covered by first responders did not vary with familiarity (Supplementary Table S4A). However, distance covered was 1.8 times higher in subsequent responders from familiar schools compared to their unfamiliar counterparts (LMM: $F_{1,23} = 9.3$, $p = 0.006$, $R^2m = 0.31$, $R^2m = 0.57$; Fig. 4; familiar: 33.3 ± 1.6 mm, unfamiliar: 18.4 ± 1.8 mm, mean ± s.e.), highlighting that the average distance moved during stages 1 and 2 improved substantially in familiar subsequent responders. There was also a significant familiarity*responder number interaction (LMM: $F_{1,114} = 5.1$, $p = 0.03$), as the magnitude of the difference between familiar and unfamiliar fish increased with the responder number. Distance covered also declined as stimulus distance increased, though this effect was not significant ($F_{1,122} = 3.72$, $p = 0.06$; Fig. S3B).

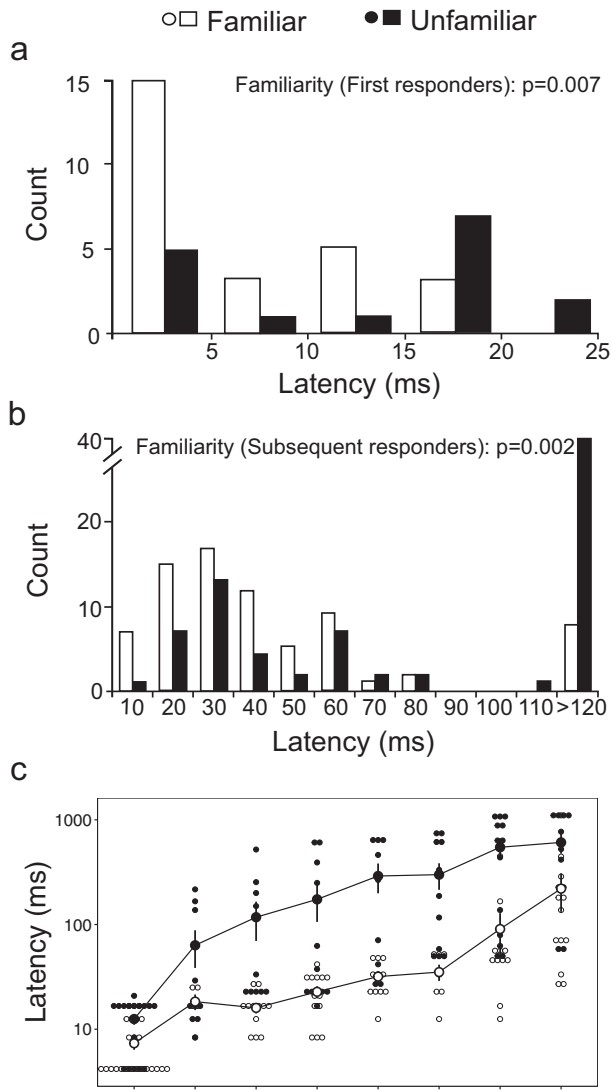

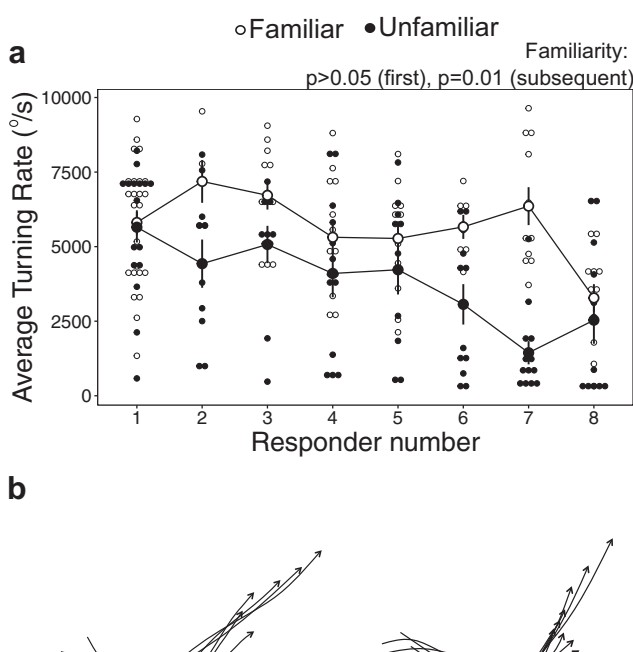

**Fig. 3 The average turning rate of the fast-start escape response in familiar versus unfamiliar schools of the damselfish *Chromis viridis*. a** Average turning rate by responder number (which denotes the sequential order of responses in each fish's respective group). Each larger dot represents the mean ± s.e., with small dots indicative of individual data points. *p*-values indicate results of linear mixed-effects model analyses (*n* = 24 schools, composed of 8 fish each). **b** Typical escape responses in the familiar (left) and unfamiliar (right) treatments. Lines represent the fish midline and arrows indicate the location of the head in successive frames at 4.2 ms intervals. The greater number of lines in the unfamiliar example indicates that the individual took more time to achieve a similar evasive manoeuvre to the individual in the familiar treatment.

**Fig. 2 The latency of the fast-start escape response in familiar versus unfamiliar schools of the damselfish *Chromis viridis*.** Frequency histograms of response latencies for first responders (**a**) and subsequent responders (**b**). **c** Response latencies by responder number, the sequential order of latencies in each fish's respective group. Each larger dot represents the mean ± s.e. (*y* axis logged for illustration), with small dots indicative of individual data points. *p*-values indicate results of the generalised linear model (**a**) and linear mixed-effects model (**b**, **c**) analyses (*n* = 24 schools, composed of 8 fish each).

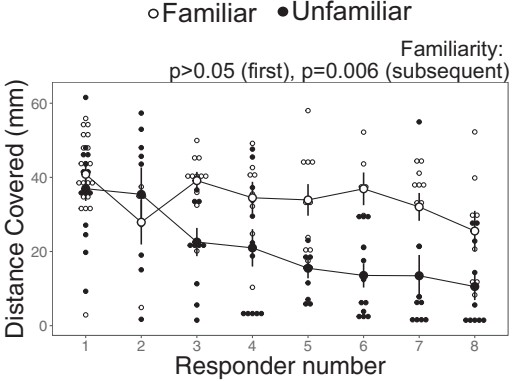

**Fig. 4 The distance covered during the fast-start escape response in familiar versus unfamiliar schools of the damselfish *Chromis viridis*.** Distance covered (i.e. the distance moved during the first 42 ms of the reaction, the mean time period for individuals to complete two body bends known as stages 1 and 2) by responder number (the sequential order of responses in each fish's respective group). This trait is indicative of the escape response's speed and acceleration (*n* = 24 schools, composed of 8 fish each). Each larger dot represents the mean ± s.e., with small dots indicative of individual data points.

**School behaviour after stimulation**. School cohesion and coordination were measured during the 100 ms time period following the simulated predator attack. This short time frame following a predator attack is crucial for survival[30,31]. School cohesion was assessed using nearest neighbour distance (NND; distance to the closest neighbour) and school area (horizontal spread of the school), while school alignment (length of the mean circular vector, a measure of the variation in fish orientations) was used as a proxy for spatial coordination of individuals within the school[56]. These school characteristics were measured at set time points throughout the escape response (time = 0, ~20, and ~100 ms after stimulus onset), representative of before, during, and after the escape response on average in this species[54] (complete model output detailed in Supplementary Table S5).

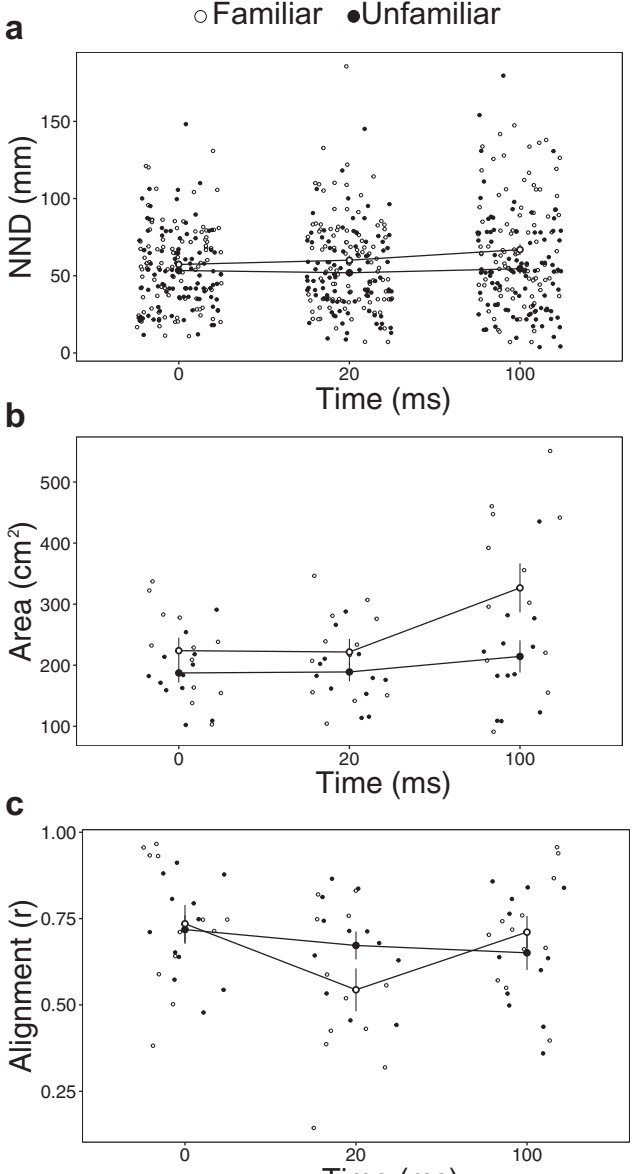

**Fig. 5 Effect of familiarity on escape performance of fish schools (Chromis viridis). a** Nearest neighbour distance (NND) denotes the distance to the closest neighbour (mm). **b** School area indicates the school's horizontal spread (cm²). **c** Alignment is a measure of the variation in the orientation of all school members. This variable is characterised by the length of the mean circular vector ($r$). Each larger dot represents the mean ± s.e., with small dots indicative of individual data points ($n =$ 24 schools, composed of 8 fish each).

Cohesion and coordination varied through time depending on familiarity treatment. Time had a significant effect on the school area ($F_{2,44} = 5.0$, $p = 0.01$) and alignment ($F_{2,44} = 5.5$, $p = 0.008$). Marginal trends for greater cohesion in familiar groups were revealed for both NND ($F_{1,22} = 3.8$, $p = 0.06$) and school area ($F_{2,44} = 3.20$, $p = 0.09$). Alignment exhibited a significant interaction between familiarity treatment and time ($F_{2,44} = 3.7$, $p = 0.03$). Familiar and unfamiliar schools all exhibited comparable NND (Tukey's Test: $p_{unfam0-fam0} > 0.05$; Fig. 5a), school area (Tukey's Test: $p_{unfam0-fam0} > 0.05$; Fig. 5b), and alignment (Tukey's Test: $p_{unfam0-fam0} > 0.05$; Fig. 5c) immediately prior to stimulation (i.e. at time = 0). However, cohesion in familiar

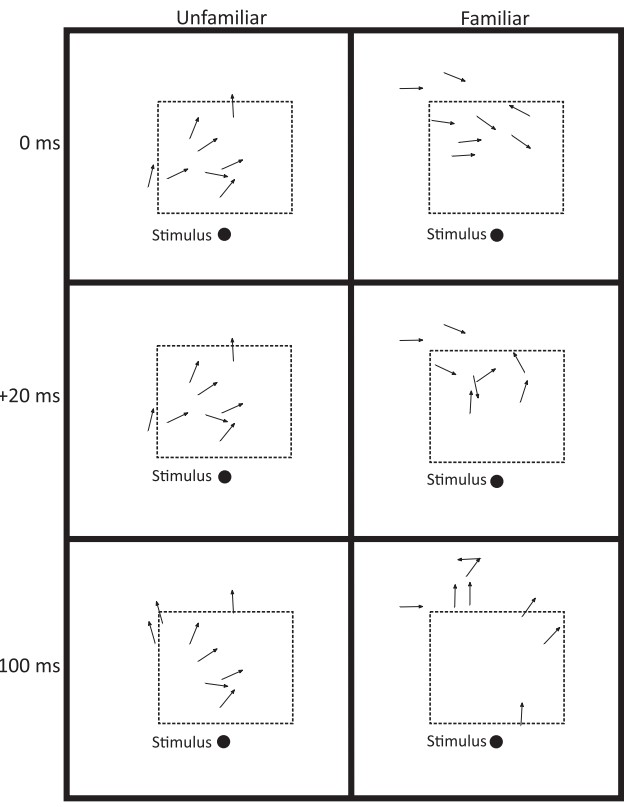

**Fig. 6 Example school behaviour after stimulation.** Each arrow shows the position and orientation of one fish. Successive frames from top to bottom illustrate the school's cohesion and alignment at 0 ms (the moment immediately before stimulation), +20 ms (during the escape response), and +100 ms (following the escape response) post-stimulation. The experimental arena measured 50 cm (long) × 40 cm (wide), with directional flow from right to left in the illustration. Both illustrated schools were stimulated by the right lateral stimulus, with an identical lateral stimulus on the left side of the arena, relative to the directional flow (both 2 cm from each of the lateral walls in the centre of the swim tunnel). To control for a stimulus side preference, the use of the left or right lateral stimulus was alternated between trials.

schools decreased at $t = 20$ ms and $t = 100$ ms following the simulated predator stimulus, with school area rising by 46% on average in familiar schools (Tukey's test: $p_{fam0<fam100} = 0.03$, $p_{fam20<fam100} = 0.02$; for all other comparisons between familiarity treatment and time, $p > 0.05$; Fig. 5b). Unfamiliar schools, conversely, maintained comparable cohesion through the 100 ms time period post-stimulation (for all comparisons, $p > 0.05$). While alignment decreased by 26% on average in familiar schools at 20 ms following the stimulus (Tukey's test: $p_{fam0>fam20} = 0.002$), unfamiliar schools remained consistent for the 100 ms period following stimulation (for all other comparisons, $p > 0.05$). In familiar schools, alignment recovered to pre-stimulus levels by 100 ms after the stimulus (Tukey's test: $p_{fam20<fam100} = 0.01$). Representative responses of a familiar and an unfamiliar school are illustrated in Fig. 6. For all mixed-effects models presented above, forest plots showing effect sizes (through the structure correlation coefficient ± 95% confidence intervals) for fixed effects are displayed in Supplementary Fig. S4.

## Discussion

The present study illustrates the importance of social context in an individual's ability to respond to threats. These results show

that the group's interaction history (e.g. degree of familiarity among group-mates) can substantially enhance individual escape performance following a predator attack, but comes at the expense of group cohesion and coordination. In first responders, the latency of familiar fish was significantly faster than unfamiliar fish, demonstrating that a familiar social context increases individual readiness to escape. Subsequent responders in familiar schools exhibited both shorter latencies and higher kinematic performance than those from unfamiliar schools, suggesting social communication and trust in the accuracy of information received from school-mates may be enriched with familiarity. Because unfamiliar fish show reduced reactivity and kinematic performance (i.e. key traits in the survival of predator attacks[30,31]), familiarity likely functionally alters survival in social animal groups in natural habitats.

For first responders, the reduced responsiveness in unfamiliar fish could be explained by a delay in the initiation of the neural networks driving the escape response. The fast-start response is typically mediated by the higher-order command neurons M-cells, which are known to generate rapid, short-latency responses to ambient predation threats[26,28,29]. Familiar fish exhibited both faster latency responses for first responders as well as a higher proportion of short latencies overall[22,28]. Interestingly, despite their slower latency to respond, unfamiliar first responders showed kinematic performance comparable to familiar first responders. This high-performance motor output in line with that of familiar fish suggests similarity in the neural control of the response to the mechanical stimulus. However, the higher cognitive demand involved in inspecting unfamiliar schooling partners likely delays initiation of the M-cell, causing a longer latency but an equally high motor output response. Thus, familiarity may be a key proximate mechanism underlying social behaviour in gregarious species[57].

Conversely, the slower latency and motor output responses typical of unfamiliar subsequent responders were comparable to those initiated by other reticulospinal neurons in the brainstem escape network[22,58–60]. This result suggests that frequently in unfamiliar social groups, the perceived threat posed by integrated direct and social stimulation did not meet the threshold necessary to trigger an M-cell response (reviewed in ref. [27]). A wealth of literature suggests that M-cell neurons exhibit plasticity in inhibition and excitation in response to a range of factors[26,61]. Here, we suggest that social context played an important modulatory role in escape performance due to a combination of behavioural and physiological mechanisms. Exposure to unfamiliar new-comers can induce a range of behavioural effects, most notably the need to increase conspecific inspection to determine the accuracy of social information and re-establish an individual's niche in the larger group[8,9]. This additional task can exhaust the cognitive process of awareness, limiting the attention available for other tasks like vigilance and defence[62]. Furthermore, the loco-motor response of "unknown" first responders may trigger reactions with lower strength in subsequent responders, compared to schools in which the first responders are familiar fish. Studies in other contexts have illustrated the effects of distraction on startle responsiveness. For example, silver-spotted sculpins involved in foraging are less reactive to predator attacks than non-foraging conspecifics[38]. This elevated cognitive demand in an unfamiliar social context could also manifest in a physiological stress response. Previous work has recorded spikes in cortisol and other glucocorticoid stress hormones following exposure to unfamiliar conspecifics[39–41,63]. Neuronal inhibition can occur in response to these stress hormones[42,61,64], which in turn can modify the stimulus threshold necessary for initiation of an escape response[65]. Future work should focus on characterising the underlying behavioural and physiological mechanisms driving

the reduction in reactivity (in both first and subsequent responders) and locomotor performance (in subsequent responders only) in an unfamiliar social context.

For subsequent responders, the effectiveness of information-sharing networks may have been enhanced through familiarity, driving the observed differences in reactivity and kinematic performance between familiar and unfamiliar fish as the responder number increased. Transmission of social cues through fish schools often propagates through individual changes in locomotion (e.g. speed and direction)[66,67], with information about risk communicated through sudden movements[23] and visual cues of neighbours' escape[25]. However, the response of neighbouring fish to first responders can vary greatly with several factors. Rosenthal et al.[25] showed that an initiator's startle response propagates more readily through highly connected neighbours than weaker networks, outweighing physical properties like group density, individual position, number of nearest neighbours, and the kinematic properties (e.g. speed and acceleration) of the initiator. In this context, familiarity may enhance the interaction network among neighbours, strengthening waves of behavioural change following an initiator's response. Neighbours aim to maximise the number of correct reactions (e.g. reacting to the presence of predators) and minimise reactions to inaccurate information (e.g. reacting to harmless stimuli)[34–36]. Thus, subsequent responders may modulate their sensitivity to social information with the degree of uncertainty about information accuracy[35].

Although group coordination and cohesion are typically thought to increase under predation threat to minimise mortality[68–71], here, we show that enhanced individual fast-start performance, as was typical in familiar fish, is associated with a temporary reduction in these traits in the first 100 ms following a predator attack. Unlike some past studies[32,72], we observed no change in school cohesion or coordination with familiarity until after the simulated predator attack, indicating that the pre-startle school structure did not dictate group-level responsiveness or performance. Following the simulated predator attack, the faster individual escape manoeuvres typical of familiar fish led to a reduction in school-level traits, while unfamiliar fish maintained group structure through a weaker response to the threat (Fig. 6). Coordination in familiar fish initially drops as the fish begin to turn in stage 1 (as evidenced by the reduction in alignment 20 ms after the stimulus), with cohesion then declining as individuals accelerate in stage 2 (illustrated by the rise in school area 100 ms post-stimulus). Reduced coordination and cohesion may be evolutionarily advantageous if these changes increase the so-called confusion effect, in which prey capture is reduced by the predator's inability to single out an individual for attack due to sensory confusion[2,73,74]. As the latency of unfamiliar fish was also substantially longer, this apparent "maintenance" of group structure could also be temporary, with unfamiliar schools potentially exhibiting the same drop in cohesion and coordination if examined over a longer time scale. Empirical studies of predator strike success on familiar and unfamiliar social groups would allow researchers to test the fitness trade-offs of these changes in school structure with time post-predator attack.

While the effects of familiarity on fast-start performance were clear in this study, we must also consider how these effects would translate into other contexts. Intrinsic factors, such as group size and phenotypic composition (e.g. body size, personality), may alter group communication and decision-making[25,36,75–77]. Extrinsic factors, such as competition for food resources[78], heterogeneity in predation pressure[79], habitat complexity[36], and water depth[80] can also shift the physical and social network structure of fish schools. As such, future work should examine how the above factors may modulate the selective advantages of familiar group composition.

Cognitive processes are central to facilitate group living in animal species[81,82]. On an individual and group level, effective social behaviour relies on acquiring and integrating information[56,83]. Here, we highlight the pivotal role that social familiarity plays in modulating a group's ability to respond to threats, potentially driven by a combination of physiological and behavioural mechanisms on individual responsiveness and group-level information-sharing networks. Introduction of unfamiliar newcomers to social groups may burden cognitive processes associated with attention and drive uncertainty in the accuracy of socially acquired information, which can hinder or delay the decision-making of individuals[9,82,84]. Social fishes, like *C. viridis*, provide valuable comparative insights into vertebrate social behaviour, given the many analogous processes guiding social cognition, brain organisation and function, and group decision-making among fish, birds, and mammals[57,82]. Hence, this study highlights the importance of familiarity and its role in collective cognition within the context of anti-predator behaviour for both individuals and groups.

## Materials and methods

This research was conducted following guidelines and regulations from the James Cook University Animal Ethics Committee (permit number A2103).

**Fish collection and maintenance**. This experiment was conducted at the Lizard Island Research Station (LIRS) in the northern Great Barrier Reef, Australia (14°40′08″S; 145°27′34″E). Distinct schools of the tropical damselfish *C. viridis* (Pomacentridae, standard length: 3.33 ± 0.02 cm, mean ± s.e.; *n* = 192 fish) were collected from different reefs (separated by 400–3000 m) in the lagoon adjacent to LIRS using hand nets, a dilute anaesthetic solution of clove oil[85] and barrier nets. Once collected, all wild schools were maintained in sensory isolation from one another (both visual and olfactory isolation) in a flow-through aquaria system at a density of ~1 fish per 2.5 L. Fish were fed to satiation twice daily with INVE Aquaculture pellets and newly hatched *Artemia* spp.

**Time to familiarity: experimental procedure**. The length of time necessary for *C. viridis* to establish familiarity was assessed prior to escape response testing, to confirm that our study species achieves familiarity in a comparable time frame to another tropical fish species, the guppy *Poecilia reticulata*[51]. Eight experimental schools composed of 15 fish each were assembled from equally unfamiliar fish, by joining 15 individuals from 15 geographically distinct wild schools (schools separated by a minimum of 100 m). To measure the development of familiarity through time, a choice test methodology was used, adapted from the protocols described in Griffiths and Magurran[51]. In this protocol, a focal fish was placed in a translucent, porous cylinder (11 cm diameter × 35 cm height) in the centre of a testing tank (90 cm length × 30 cm width × 30 cm depth). Stimulus schools were then placed at either end of the testing tank in translucent, porous bottles (11 cm diameter × 35 cm height). These stimulus schools were composed of seven fish each, chosen randomly from either the school with which the focal fish had been housed (i.e. familiar school) or an unfamiliar school. The focal fish and the two schools were allowed to acclimate for 15 min before the barrier of the focal fish was removed using a pulley system (this methodology prevented the focal fish from seeing the observer when the barrier was removed). Trials lasted 15 min each and the amount of time that the focal fish spent schooling with each group was recorded. The location of the familiar and unfamiliar schools (right or left container) was randomised between trials, to eliminate a potential side preference effect. The focal fish was considered to be schooling when it swam within approximately two body lengths (*L*, i.e. 6 cm) of either school. One individual was tested from each experimental school every 2 days for 21 days and a different individual was used for each trial. School preference was defined as the school that the focal fish spent the greatest proportion of time with.

**Escape response: experimental procedure**. Escape responses for familiar and unfamiliar groups were examined in schools composed of eight fish each (*n* = 12 schools per treatment). All experimental schools were assembled from equally unfamiliar fish, by joining 8 individuals from 8 geographically distinct wild schools, so that all schools were equally unfamiliar at the start of the experiment. Schools from the "familiar" treatment were given a period of three weeks to familiarise prior to testing. Similarly, unfamiliar schools were housed for three weeks in groups composed of eight fish, before being reassembled with unfamiliar individuals immediately prior to experimentation (<10 min prior to the start of each 4.5-h "unfamiliar" trial, as described below).

Trials were then conducted in a custom-built laminar flow tank (50 cm length × 40 cm width × 9 cm height). This device allowed schools to swim in non-turbulent conditions at a slow uniform swim speed of approximately one *L* per second (3.2 cm/s), which mimics natural flow speed conditions on the LIRS reefs on a calm day[50]. Seawater in the system was maintained at the ambient temperature for the study period (27–29 °C).

Experimental schools were placed in the swim tunnel and allowed to acclimate for a period of 4 h. Escape responses were then elicited using a standardised threat protocol in which a mechanical stimulus was dropped from above[53]. This stimulus was a black cylindrical object (2.5 cm diameter × 12 cm length, 37.0 g) with a tapered end (to minimise surface waves), suspended 137 cm over the surface of the water in the swim tunnel. To ensure that stimulation occurred precisely when the stimulus hit the water surface, the object was released through a white PVC pipe ending 3 cm above the water's surface that prevented visual stimulation as the object dropped[86,87]. A thread connecting the stimulus to the release point prevented it from touching the bottom of the tank[86,87]. As previous studies suggest that the school's alignment during an escape response is highest with lateral stimulation at an angle of 30°–120°[56], identical stimuli were placed 2 cm from each of the lateral walls in the centre of the swim tunnel. To control for a stimulus side preference, the use of the left or right lateral stimulus was alternated between trials. These stimuli remained suspended above the swim tunnel for the duration of the acclimation period using an electromagnet. Following the acclimation period, the stimulus was released using a switch, once a minimum of 6 of the 8 fish were >3.5 cm from any wall of the swim tunnel and within four *L* of the stimulus. This criterion aided in reducing the constraining effects that the walls of the swim tunnel may exert on individuals' escape response and variable effects that stimulus distance can have on the motor output of the escape response[52]. Each school's escape response was video recorded from below at high speed (240 fps; Casio Exilim HS EX-ZR1000), using a mirror placed at a 45° angle. The swim tunnel was illuminated from above using two 500-W spotlights.

**Kinematic analysis**. Videos were analysed manually frame-by-frame using the ImageJ software and all video analysis was conducted blindly such that the researcher did not know the trial's treatment during its video analysis. The stimulus onset was defined as the frame at which the stimulus first touched the water surface. A number of individual performance characteristics and the escape response of each school were examined as defined below.

**Individual fast-start escape performance**. Individual fast-start escape performance was characterised through traits associated with reaction timing and kinematic performance. First, we examined latency, which indicates the time period between the stimulus onset (i.e. first frame where the stimulus is observed breaking the water surface) and the fish's first head movement following the stimulus in each individual fish (Fig. 1a). Individuals were assigned a responder number (as described in the results section above) based on their sequential order of reaction in their respective groups. Given that our high-speed video framing rate was 240 fps with frames 4.17 ms apart, fish tied for latency are highly unlikely to have startled each other, as we did not observe any latencies <4.17 ms (which is in line with previous work[27]). Therefore, two (or more) fish tied for the first responder were both directly startled by the stimulus and variation in the latency of first responders with familiarity would be indicative of direct effects of familiarity on individual latency performance. If subsequent responders vary according to familiarity treatment, this effect indicates a combination of effects on individual latency performance and/or within-school information transfer after the stimulus onset. This effect is because subsequent responders reacted after both the stimulus and the reaction of first responders, and therefore their response may have been influenced by both.

As a proxy for agility, we measured the average turning rate, which is calculated by dividing the stage 1 angle (Θ, the angle between the lines intersecting the head and centre of mass when stretched straight (CM) at the start and end of stage 1) by the duration of stage 1 (Fig. 1b). The location of the CM in the video footage was measured as 0.35*L* posterior of the snout, based on previous measurements of generalist fishes[88]. Stage 1 is the stage immediately following the stimulus, in which fish contract the muscles on the side of the body opposite to the stimulus, causing the fish to bend into a C shape[21]. To calculate the stage 1 angle, the *x* and *y* coordinates of the snout and centre of mass were recorded at frame 0 (i.e. one frame prior to the fish's first detectable movement following the stimulus) and in the frame at the end of stage 1 (i.e. where the maximum body angle was achieved), to determine the rotation of the line passing through these two locations on the fish's body between the beginning of the response and the end of stage 1. The stage 1 duration was determined through the number of frames taken to achieve the stage 1 angle and multiplied by 4.17 ms (the duration of time between frames, based on the filming rate of 240 fps).

As a proxy for propulsive performance, we examined distance covered, which is the distance that the fish's CM travelled within the first 10 frames (e.g. 42 ms) of their reaction (as measured as the straight line between the *x* and *y* coordinates of the CM in frame 0 and +10 frames, respectively; Fig. 1c). This time frame was determined by calculating the average duration of stages 1 and 2 in the escape response for 24 individuals (one random fish per trial). Previous work has shown that these two stages are crucial for predator avoidance[30]. In addition, the distance covered during this short time frame can be used as a proxy for swimming speed to avoid issues with wall effects. Individuals less than two *L* from any wall of the swim

tunnel at the time of their response were excluded from this analysis (10% of total[52]).

For both measures of kinematic performance (average turning rate and distance covered), we examined variation across responder numbers to inform the role of familiarity in individual performance versus individuals responding to integrated stimulation (i.e. responders 2–8 could have responded to a direct and/or social stimulation). In addition, as previous studies indicate that the stimulus distance (distance from the stimulus to the fish's CM) can influence all of the above traits (latency, average turning rate and distance covered)[24], this measure was included as a covariate in the analyses. Non-responders (3% of total fish) were assigned the highest latency value measured (1104 ms) and the lowest average turning rate and distance covered measured (322.59°/s and 1.477 cm, respectively).

**School behaviour**. School escape response was characterised through school cohesion (nearest neighbour distance and total school area) and spatial organisation (alignment of individuals). These school characteristics were measured at set time points throughout the escape response (0, 20.8 and 99.8 ms following the stimulus)[56], as this short time frame following a predator attack is crucial for survival[30,31] and previous studies indicate that these time points represent the school's pre-stimulation behaviour as well as the latencies for 33% and 66% of *C. viridis* individuals, respectively[54].

(1) Nearest neighbour distance (NND): distance to the closest neighbour within the school, as measured by the distance from each fish's CM.
(2) School area: the school's horizontal spread (cm$^2$), as measured by the area between the fish at the edge of the school (defined as those fish whose head was at the vertex of the smallest convex polygon encompassing the group)[89].
(3) Alignment: the variation in the orientation of all school members to the horizontal (corresponding to the direction of flow; 0° = facing into the flow towards the front of the tank, 180° = oriented with the flow towards the back of the tank). As alignment angles spanned up to 360°, circular statistics were employed to determine the mean, known as a circular vector. The variability around this mean was then calculated, and is inversely related to the length of the mean circular vector ($r$)[90], which spans from 0 (indicating the angles are random) to 1 (indicating all angles were identical). The mean circular vector $r$ was calculated in the software Oriana 4.

**Statistics and reproducibility**. All statistical analysis was conducted in the R Statistical Environment (v3.2.4,[91]), using generalised linear models (GLM) and linear mixed-effects models (LMM) in the packages "lme4", "car", "MuMIn", "multcomp", "partR2", "patchwork", "ltm", and "ggplot2". Residual and quantile-quantile plots were assessed visually for each model to ensure that all assumptions were met. A Shapiro-Wilk test and Bartlett test of the model residuals were also used to confirm the assumptions of normality and homogeneity of variance, respectively. Log and box cox transformations were used as needed to meet these assumptions. The time to familiarity was analysed using a GLM, in which time to familiarity was analysed using a GLM with a binomial distribution, comparing preference for the familiar school (true or false) by day and school. All individual performance traits were analysed using separate models for "first responders" and "subsequent responders". First, analyses examined the latency, average turning rate, and distance covered by first responders to assess direct reactions to the stimulus (i.e. not impacted by social stimulation). The latency of first responders was analysed using a GLM, with school type, stimulus distance, and their interaction as explanatory variables, weighted for the frequency of responders that tied for the first responder position in each group. The average turning rate and distance covered of first responders was assessed using LMMs with school type, stimulus distance, and their interaction as fixed effects, and school as a random effect (with this random effect included to account for the tied first responders in many schools; note that these tied responders varied in their kinematic output unlike their tied latency). For subsequent responders, latency, average turning rate and distance covered were all analysed using LMMs with school type (familiar or unfamiliar) and responder number (2–8) as fixed effects, stimulus distance as a covariate and school as a random effect. The average turning rate of subsequent responders was log-transformed to meet the model's assumption of homoscedasticity. All relevant interactions were also included in each model (among school type, responder number and stimulus distance). In addition, as previous work (using microelectrodes to measure M-cell action potentials) conducted using a comparable temperature and mechanical stimulus found an average latency of 16 ms in M-cell fast-start escape responses[22], we also converted latency to a binomial variable. Here, we defined "fast" responses as exhibiting a latency <16.7 ms, while "slow" responses were defined as >16.7 ms. We used this time frame, as it was the closest we could get to 16 ms given our video recording frame rate (as frames were separated by 4.17 ms at our 240 fps frame rate). Differences in the fast and slow responses between familiar and unfamiliar schools were assessed using a generalised linear mixed-effects model with a binomial distribution, with school type as a fixed effect and school as a random effect. Differences in NND, school area and alignment were assessed using LMMs, with familiarity treatment and time as fixed effects and school as a random effect. For NND, the individual was also included as a random effect (as this variable was assessed on an individual basis). No auto-correlation correction was deemed necessary for these repeated measures, based on

the Akaike information criterion (AIC) for models with and without this correction. Tukey's HSD post hoc tests were used to further investigate the differences detected. The $R^2$ for all models is detailed in the results above (including the R$^2$m and R$^2$c, representing the variance explained by the fixed effects only and the fixed effects plus random effects, respectively). In addition, the models' effect sizes are included in the results section above (as represented by the point biserial correlation coefficient for the GLMs) or graphically in the supplementary material (represented by the structure coefficients for each fixed effect).

**Reporting summary**. Further information on research design is available in the Nature Research Reporting Summary linked to this article.

## Data availability
All data and code are available through the NSUWorks Data Repository (https://nsuworks.nova.edu/occ_facdatasets/13/).

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

## Acknowledgements

We thank the Lizard Island Research Station staff, R. Barrett, K. Corkill, R. Zemoi, B. Allan, M. Palacios and S. Brown for logistical support, R. Jones for statistical advice, and Ø. Øverli for providing comments on an earlier version of this manuscript. Funding was provided by an Australian Postgraduate Award, International Postgraduate Research Scholarship, Lizard Island Reef Research Foundation Doctoral Fellowship, Great Barrier Reef Marine Park Authority Science for Management Award and James Cook University Graduate Research Scheme to L.E.N. and an Australian Research Council Centre of Excellence for Coral Reef Studies grant to M.I.M. (EI140100117).

## Author contributions

L.E.N, M.I.M., J.L.J. and P.D. conceived the study idea and designed the experiments. L.E.N. conducted the experiments. L.E.N. completed the video analysis, with input from P.D. L.E.N completed the statistical analysis, with input from J.L.J. L.E.N. wrote the initial draft of the manuscript, and all authors contributed to editing the final manuscript.

## Competing interests

The authors declare no competing interests.
