## [Peer Review File · Communications Biology]

Reviewers' Comments:

Reviewer #1:

Remarks to the Author:

In this paper, the authors test if prior familiarity with shoal-mates changes the response time and kinetics of small social groups of tropical damselfish *Chromis viridis* in a simulated predator condition. The authors provide results to show that the latency to response decreases, while, the social group's cohesion decreases with familiarity. Authors also show that the first responders' kinematic performance in unfamiliar groups is similar to that of the first responders in familiar groups in spite of the increase in latency to, thus suggesting recruitment of the same neural networks and circuits engaged in propulsion. The authors interpret these results to demonstrate that the stress of unfamiliarity causes such deficits and is overcome by enduring social bonds.

In general, I find the study very interesting, the methods applied adequate, and the results compelling to present to other scientists, and I am supportive of the publication of the study at Comm. Biol.

However, I do have a few points that I would like the authors to address.

1. I find the author's interpretation of attributing the change in the latency to "increased stress" of interacting with unfamiliar individuals somewhat unsatisfactory. As the term "stress" carries a particular connotation in the literature (especially with the abundance of rodent studies), I suggest the authors reconsider the terminology and the presentation of interpretation in the discussion. As the authors expand a little in the subsequent lines of discussion regarding what they may mean by stress, this becomes somewhat clearer, but I suggest a revisit/re-write of the abstract, intro, and discussion to present a coherent argument. What they mean is the higher cognitive demands of unfamiliar shoaling partners possibly reduces the use of public information for decision making in these populations, as the history of interactions has not yet been established. That is, individuals may be relying primarily on private information rather than public information of established schools. The ideas of social cognition and such possibility have been discussed extensively in the literature even if not for this species in particular (see 1. Bshary R, Gignis S, Vail AL: Social cognition in fishes. Trends Cogn. Sci. 2014, 18:465-471. and 2. Pouca, C. V., sciences, C. B. C. O. I. B., 2017. <https://doi.org/10.1016/j.cobeha.2017.03.002>) among many others, so I don't see the problem in making a more direct interpretation.

2. I am not entirely convinced that the data presented demonstrate "enduring social bonds" in any clear manner. This is a peculiar choice of phrase as the methods employed in fact suggest that familiarity in this species is established within 3 weeks, that is, new connections with unfamiliar individuals can be established rapidly (When forced)! That seems like relatively small period of time (relative to the life span of this species) to categorize as enduring social bonding. In the form of questions, the authors may wish to consider -

a) how was long-lasting social bonds demonstrated by this experiment? The results show that familiarity with shoal members improves reaction time.

b) What can be considered an enduring bonding period for this species? Do the authors mean social recognition and memory of individuals? Enduring social bonds are more appropriate for demonstrations that show individual recognition based action. Was this demonstrated?

3. In the abstract; What are higher-level animals? Animals with larger brains? Is Octopus considered a lower level or a higher level animal?

4. Line 70 of the merged document; "However, the role of social familiarity in modulating the timing and kinematic performance of individual evasive manoeuvres, and whether these effects are driven by changes in individual reactivity or social communication, has yet to be investigated." There have been several previous attempts in the direction. Eg.

<https://www.pnas.org/content/112/15/4690.short> Can the authors help specify if the specific type of analysis their study provides (perhaps for this species) distinguishes it from other studies? OR, perhaps it improves on, or adds to them. Additional notes in the introduction will be helpful.

5. Line 136 " Specifically, we found that the first responders in familiar groups exhibit latencies that

are approximately 60% shorter than first responders in unfamiliar groups (Generalized Linear Model (GLM) $F_{1,22} = 9.2$, $p = 0.007$, $R^2 = 0.43$; Fig. 2A), and are hence reacting more quickly to the predator stimulus (familiar: 7.5 ± 0.9 ms, unfamiliar: 12.5 ± 1.5 ms, mean \pm s.e.)."

Can the authors also report the size of the effect in the difference in response? This can be done in a number of different ways, the simplest being providing numerical and graphical data using estimation stats (www.estimationstats.com). Please see this and other discussions on including effect size doi: <https://doi.org/10.1038/d41586-019-00857-9>

6. Methods and statistics related

1. Regarding Kinematic analysis How was "first reaction" scored? Manually, frame by frame by the experimenter? Also, unclear the method used to calculate stage 1 angle (θ)? Was it done using a script in imageJ

2. Can the authors describe the entire method of imageJ analysis? How were individuals tracked? Can they upload sample videos analysis scripts (if any) and output? ImageJ use details - were macros used, etc. could not be verified.

3. A simple video of the entire analysis pipeline may be necessary to evaluate the methodology used.

4. Explicit statistical statements reported as confirmed in the reporting summary are missing in the manuscript text file. Each point checked as "confirmed" should be explicitly stated in the Methods section in the statistical analysis section.

Reviewer #2:

Remarks to the Author:

In this study the authors tested whether familiarity among group mates alters the speed with which individual damselfish within a group respond to a simulated predator attack. They used groups of fish that had either lived together for 3 weeks or had just been introduced. They found that in familiar groups of damselfish, the first fish to respond to the startle was faster to respond, and that subsequently responding fish were also faster. The first responder in both familiar and unfamiliar groups covered similar distances due to their fast-start, but subsequent responders in familiar groups had longer distances than individuals in unfamiliar groups. Finally, familiar groups appeared to respond in a less cohesive way, than did un-familiar groups. Altogether, I found the experimental design clean, I think the data were appropriately analyzed and the conclusions drawn from them well-supported. My main comment has to do with how they set up their motivation for this study. I think improving this aspect of the manuscript will help convince readers why this paper is interesting and so increase its impact.

Main comment:

1 – Introduction. Right now, in the introduction the study is basically motivated as "familiarity with group mates is often a good thing, so let's see whether it affects fast-start escape responses." As a reader, I really wanted to see why would we expect to familiarity to affect escape responses in the first place and then just as importantly, HOW would this effect occur?

Each paragraph in the intro ends with a line that is essentially the same – the "role of social group composition in driving ...responses to predation threat remains poorly understood." (Line 36). But that is not a very compelling motivation – maybe it remains poorly understood because there is no strong reason to assume that familiarity would affect response to predation threat! But I don't think that is actually the case. Rather I think there are very compelling reasons to think that familiarity would/could affect escape times. The authors allude to these reasons very briefly in the intro by saying things like "the mechanisms driving [the effects of familiarity on escape responses] remain unclear." But then don't really do into detail as to what those mechanisms may be other than saying that M-cells might be involved.

Then as I read on, I realized that actually, all the motivation for the study that I was looking for in the introduction, is already in the discussion! For example, the paragraph that starts on Line 218 discusses how (un)familiarity could affect the initiation of neural network responses and potential differences between M-cells and other neurons. They also suggest on Line 248 that unfamiliarity

might cause stress which can have a neural inhibition effect! They also suggest on Line 262 that familiarity might influence interaction networks! So here were all the compelling reasons WHY and HOW we would expect familiarity to affect something like escape responses. These reasons need to be moved to the introduction. I understand that the authors may not be able to fully definitively test each of these possible mechanisms, but alluding to them as possibilities provides a much more convincing link between familiarity and escape responses.

Along those same lines, I think a little bit extra detail about the difference between M-cells and 'other reticulospinal neurons' might be useful for folks that aren't familiar (like me). It seems that you can tell something about which type of neuron generated the fast-start response based on the response time? More detail on that would be useful.

Minor comments:

Line 14: This is maybe a knee-jerk reaction, but I don't think that there is any need to suggest that fish are not a "higher-level" animal. What exactly is a higher-level animal anyway? Additionally, there are well known examples of 'enduring social bonds' in many fish species (thinking about cooperative breeding cichlids, for example).

Line 57: When I first read this, I was wondering why on earth the authors needed to specifically mention M-cells here as they were never mentioned again, until the discussion. But clearly, the authors wanted to mention them because apparently you can tell whether they initiate the response based on the response time. Another sentence here outlining that (responses generated by M-cell are generally performed in under 16ms whereas responses generated by other neurons take longer) would be useful.

Line 61: What kind of mechanisms??? (this is now addressed in my main comment)

Line 70: What is the mechanism here that you would expect something like familiarity to affect kinematic performance?? (this is now addressed in my main comment)

Line 186 and throughout: I always appreciate when authors don't just state that there is an effect of X factor, but also state in what direction this effect is.

Line 207: Not sure I quite understand "homogeneity in the level of familiarity amongst group-mates" – aren't both types of groups (familiar and unfamiliar) "homogenous" in their level of familiarity, in that they are all either very familiar with each other, or not familiar at all? The reference to homogeneity makes me think that some individuals in the group are more/less familiar with each other than others.

Methods: I'll just note here that I found their methods well described and easy to follow, especially their statistical analyses (which I normally am very nit-picky about). Just one quick question – on Line 441 the authors state that they include school as a random effect in their models testing whether turning rate and distance covered of first responders differed across school types. But the response variable is of first responders, so shouldn't there only be one value per school and hence no need to include school as a random effect?

Reviewer #1:

*In this paper, the authors test if prior familiarity with shoal-mates changes the response time and kinetics of small social groups of tropical damselfish *Chromis viridis* in a simulated predator condition. The authors provide results to show that the latency to response decreases, while, the social group's cohesion decreases with familiarity. Authors also show that the first responders' kinematic performance in unfamiliar groups is similar to that of the first responders in familiar groups in spite of the increase in latency to, thus suggesting recruitment of the same neural networks and circuits engaged in propulsion. The authors interpret these results to demonstrate that the stress of unfamiliarity causes such deficits and is overcome by enduring social bonds.*

In general, I find the study very interesting, the methods applied adequate, and the results compelling to present to other scientists, and I am supportive of the publication of the study at Comm. Biol. However, I do have a few points that I would like the authors to address.

1. I find the author's interpretation of attributing the change in the latency to "increased stress" of interacting with unfamiliar individuals somewhat unsatisfactory. As the term "stress" carries a particular connotation in the literature (especially with the abundance of rodent studies), I suggest the authors reconsider the terminology and the presentation of interpretation in the discussion.

As the authors expand a little in the subsequent lines of discussion regarding what they may mean by stress, this becomes somewhat clearer, but I suggest a revisit/re-write of the abstract, intro, and discussion to present a coherent argument. What they mean is the higher cognitive demands of unfamiliar shoaling partners possibly reduces the use of public information for decision making in these populations, as the history of interactions has not yet been established. That is, individuals may be relying primarily on private information rather than public information of established schools. The ideas of social cognition and such possibility have been discussed extensively in the literature even if not for this species in particular (see 1. Bshary R, Gingins S, Vail AL: Social cognition in fishes. Trends Cogn. Sci. 2014, 18:465-471. and 2. Pouca, C. V., sciences, C. B. C. O. I. B., 2017. <https://doi.org/10.1016/j.cobeha.2017.03.002>) among many others, so I don't see the problem in making a more direct interpretation.

We agree and have made sure that the terminology related to "stress" is clarified throughout the manuscript, so that it is clear that the term "stress" refers to a physiological response related to stress hormones like cortisol and use alternative terminology throughout (other than stress) to indicate circumstances in which processes like cognition are being further taxed by the additional task of conspecific inspection. One key section where we have made these changes is the discussion paragraph starting on line 259, in which we have modified the language extensively to clarify the above points related to stress versus cognition. We have also added in references to Bshary et al. 2014 and Pouca and Brown 2017 throughout the discussion to support our arguments related to social cognition.

2. I am not entirely convinced that the data presented demonstrate "enduring social bonds" in any clear manner. This is a peculiar choice of phrase as the methods employed in fact suggest that familiarity in this species is established within 3 weeks, that is, new connections with unfamiliar individuals can be

established rapidly (When forced)! That seems like relatively small period of time (relative to the life span of this species) to categorize as enduring social bonding. In the form of questions, the authors may wish to consider -

a) how was long-lasting social bonds demonstrated by this experiment? The results show that familiarity with shoal members improves reaction time.

b) What can be considered an enduring bonding period for this species? Do the authors mean social recognition and memory of individuals? Enduring social bonds are more appropriate for demonstrations that show individual recognition based action. Was this demonstrated?

We agree with your interpretation of the phrase “enduring social bonds” and have changed the terminology throughout the manuscript to reflect this issue. You are correct that it could be considered subjective to label an association generated in weeks to an enduring social bond when this species is known to live up to several years in the wild. One example where this terminology was altered is the last sentence of the abstract, which now reads:

“While social memory is often attributed to mammals with advanced cognitive abilities, these findings suggest similar relationships can dictate behaviour in other vertebrates (e.g. fishes), with fitness implications for individuals and groups.” (lines 32-34)

3. In the abstract; What are higher-level animals? Animals with larger brains? Is Octopus considered a lower level or a higher level animal?

This is a very good point, and one mentioned by reviewer 2 as well. This phrasing referred to animals with advanced cognitive abilities, like birds and humans, but we can see based on your comment that this terminology could easily be misconstrued. Hence, we have modified the last sentence of the abstract and discussion to better represent our intended meaning, which now say:

Abstract (lines 32-34): “While social memory is often attributed to mammals with advanced cognitive abilities, these findings suggest similar relationships can dictate behaviour in other vertebrates (e.g. fishes), with fitness implications for individuals and groups.”

Discussion: (lines 335-340): “Social fishes, like *C. viridis*, provide valuable comparative insights into vertebrate social behaviour, given the many analogous processes guiding social cognition, brain organization and function, and group decision-making among fish, birds, and mammals. Hence, this study highlights the importance of familiarity and its role in collective cognition within the context of antipredator behaviour for both individuals and groups.”

4. Line 70 of the merged document; “However, the role of social familiarity in modulating the timing and kinematic performance of individual evasive manoeuvres, and whether these effects are driven by changes in individual reactivity or social communication, has yet to be investigated.” There have been several previous attempts in the direction. Eg. <https://www.pnas.org/content/112/15/4690.short> Can the authors help specify if the specific type of analysis their study provides (perhaps for this species) distinguishes it from other studies? OR, perhaps it improves on, or adds to them. Additional notes in the introduction will be helpful.

We have adjusted the language in the introduction to make more clear how we are building on this previous body of work, and where the gaps remain in the literature on this topic (see lines 74-78, which say: “Despite the abundant evidence for the benefits of familiarity to processes including defence, there

remains a gap in our understanding about the mechanisms driving these effects, and, particularly, whether they stem from changes in individual behaviour, social communication, or some combination of individual and social factors.”).

5.Line 136 “ Specifically, we found that the first responders in familiar groups exhibit latencies that are approximately 60% shorter than first responders in unfamiliar groups (Generalized Linear Model (GLM) $F_{1,22} = 9.2$, $p = 0.007$, $R^2 = 0.43$; Fig. 2A), and are hence reacting more quickly to the predator stimulus (familiar: 7.5 ± 0.9 ms, unfamiliar: 12.5 ± 1.5 ms, mean \pm s.e.).”

Can the authors add also report the size of the effect in the difference in response? This can be done in a number of different ways, the simplest being providing numerical and graphical data using estimation stats (www.estimationstats.com). Please see this and other discussions on including effect size doi: <https://doi.org/10.1038/d41586-019-00857-9>

We agree that providing effect sizes is important to understand the amount of variation explained by a model. In the previously submitted version of the manuscript, we included information on each model’s R^2 (including the marginal and conditional R^2 for mixed-effects models, to indicate the variance explained by the fixed effects only and the fixed effects plus the random effects, respectively; these definitions now provided in the methods section, lines 507-509). In addition, in our revised manuscript, we have added in additional information on correlation coefficients. We added the point biserial correlation coefficient for the relationship between latency of first responders and school type (using R library “lrm”; line 163). To ensure that all other correlation coefficients accurately reflected the variation induced by random effects in mixed-effects models, we have illustrated the structure correlation coefficient for each fixed effect investigated in all other models (and their 95% confidence intervals) in forest plots (which show the effect sizes graphically with their 95% confidence intervals) in the supplementary material (using R libraries “partR2” and “patchwork”), following the protocol outlined in the following manuscript:

Stoffel MA, Nakagawa S, Schielzeth H (2021) partR2: Partitioning R^2 in generalized linear mixed models. *bioRxiv* doi: <https://doi.org/10.1101/2020.07.26.221168>

6. Methods and statistics related

1. Regarding Kinematic analysis How was "first reaction" scored? Manually, frame by frame by the experimenter? Also, unclear the method used to calculate stage 1 angle (Θ)? Was it done using a script in imageJ

2. Can the authors describe the entire method of imageJ analysis? How were individuals tracked? Can they upload sample videos analysis scripts (if any) and output? ImageJ use details - were macros used, etc. could not be verified.

3. A simple video of the entire analysis pipeline may be necessary to evaluate the methodology used.

4. Explicit statistical statements reported as confirmed in the reporting summary are missing in the manuscript text file. Each point checked as "confirmed" should be explicitly stated in the Methods section in the statistical analysis section.

ImageJ analyses were conducted manually and blinded (without a Macro or automated tracking program) in ImageJ as many automated tracking software packages struggle to maintain tracking when several animals are moving simultaneously. This has been clarified in the methods now (lines 396-398). Additional details related to how the stage 1 angle and duration and distance covered were quantified are now provided in the methods section as well (lines 420-442). Based on your comment above, effect

sizes are now illustrated in the supplementary material through forest plots illustrating the structure correlation coefficients and their 95% confidence intervals, as well as the point biserial correlation coefficient for the relationship between latency of first responders and school type. In addition, we have changed the title for our statistics subsection in the **Materials and Methods** to “Statistics and Reproducibility” in line with the guidelines in the *Communications Biology* checklist.

Reviewer #2:

In this study the authors tested whether familiarity among group mates alters the speed with which individual damselfish within a group respond to a simulated predator attack. They used groups of fish that had either lived together for 3 weeks or had just been introduced. They found that in familiar groups of damselfish, the first fish to respond to the startle was faster to respond, and that subsequently responding fish were also faster. The first responder in both familiar and unfamiliar groups covered similar distances due to their fast-start, but subsequent responders in familiar groups had longer distances than individuals in unfamiliar groups. Finally, familiar groups appeared to respond in a less cohesive way, than did un-familiar groups. Altogether, I found the experimental design clean, I think the data were appropriately analyzed and the conclusions drawn from them well-supported. My main comment has to do with how they set up their motivation for this study. I think improving this aspect of the manuscript will help convince readers why this paper is interesting and so increase its impact.

Main comment:

1 – Introduction. Right now, in the introduction the study is basically motivated as “familiarity with group mates is often a good thing, so let’s see whether it affects fast-start escape responses.” As a reader, I really wanted to see why would we expect to familiarity to affect escape responses in the first place and then just as importantly, HOW would this effect occur?

Many thanks for this comment and the associated comments below. We agree that highlighting the theoretical underpinning for this work increases its potential impact. We have added in two paragraphs to the introduction to address this point, addressing the social and individual mechanisms that may drive the hypothesized effects. (lines 79-96)

Each paragraph in the intro ends with a line that is essentially the same – the “role of social group composition in driving ...responses to predation threat remains poorly understood.” (Line 36). But that is not a very compelling motivation – maybe it remains poorly understood because there is no strong reason to assume that familiarity would affect response to predation threat! But I don’t think that is actually the case. Rather I think there are very compelling reasons to think that familiarity would/could affect escape times. The authors allude to these reasons very briefly in the intro by saying things like “the mechanisms driving [the effects of familiarity on escape responses] remain unclear.” But then don’t really do into detail as to what those mechanisms may be other than saying that M-cells might be

involved.

To avoid this repetitive structure and make the study motivation clearer, we have deleted the earlier knowledge gap statements in the introduction, instead having the introduction culminate in a more specific knowledge gap, prior to describing the potential mechanisms explaining how these effects occur (see lines 74-78, which say: “Despite the abundant evidence for the benefits of familiarity to processes including defence, there remains a gap in our understanding about the mechanisms driving these effects, and, particularly, whether they stem from changes in individual behaviour, social communication, or some combination of individual and social factors.”).

Then as I read on, I realized that actually, all the motivation for the study that I was looking for in the introduction, is already in the discussion! For example, the paragraph that starts on Line 218 discusses how (un)familiarity could affect the initiation of neural network responses and potential differences between M-cells and other neurons. They also suggest on Line 248 that unfamiliarity might cause stress which can have a neural inhibition effect! They also suggest on Line 262 that familiarity might influence interaction networks! So here were all the compelling reasons WHY and HOW we would expect familiarity to affect something like escape responses. These reasons need to be moved to the introduction. I understand that the authors may not be able to fully definitively test each of these possible mechanisms, but alluding to them as possibilities provides a much more convincing link between familiarity and escape responses.

Many thanks for your comment. For the introduction, it is always a bit tricky to find the balance between excess speculation and sufficient detail/motivation. In line with your suggestion, and mentioned above, we have added in two additional paragraphs to the introduction to better outline the potential social and individual mechanisms driving these effects. (lines 79-96)

Along those same lines, I think a little bit extra detail about the difference between M-cells and ‘other reticulospinal neurons’ might be useful for folks that aren’t familiar (like me). It seems that you can tell something about which type of neuron generated the fast-start response based on the response time? More detail on that would be useful.

Thank you for pointing this out. We have added in additional explanation of how responses differ between those initiated by Mauthner cells, and those initiated by other reticulospinal neurons. (lines 67-72)

Minor comments:

Line 14: This is maybe a knee-jerk reaction, but I don’t think that there is any need to suggest that fish are not a “higher-level” animal. What exactly is a higher-level animal anyway? Additionally, there are well known examples of ‘enduring social bonds’ in many fish species (thinking about cooperative breeding cichlids, for example).

This is a very good point, and one mentioned by reviewer 1 as well. This phrasing referred to animals with advanced cognitive abilities, like birds and humans, but we can see based on your comment that

this terminology could easily be misconstrued. Hence, we have modified the last sentence of the abstract and discussion to better represent our intended meaning, which now say:

Abstract (lines 32-34): “While social memory is often attributed to mammals with advanced cognitive abilities, these findings suggest similar relationships can dictate behaviour in other vertebrates (e.g. fishes), with fitness implications for individuals and groups.”

Discussion: (lines 335-340): “Social fishes, like *C. viridis*, provide valuable comparative insights into vertebrate social behaviour, given the many analogous processes guiding social cognition, brain organization and function, and group decision-making among fish, birds, and mammals. Hence, this study highlights the importance of familiarity and its role in collective cognition within the context of antipredator behaviour for both individuals and groups.”

Line 57: When I first read this, I was wondering why on earth the authors needed to specifically mention M-cells here as they were never mentioned again, until the discussion. But clearly, the authors wanted to mention them because apparently you can tell whether they initiate the response based on the response time. Another sentence here outlining that (responses generated by M-cell are generally performed in under 16ms whereas responses generated by other neurons take longer) would be useful.

Thanks for pointing this out. In line with your suggestion, we have specified that we classify a rapid fast-start response as one with a latency <16 ms. (line 70).

Line 61: What kind of mechanisms??? (this is now addressed in my main comment)

Yes, see response above in relation to this question.

Line 70: What is the mechanism here that you would expect something like familiarity to affect kinematic performance?? (this is now addressed in my main comment)

Yes, see response above in relation to this question.

Line 186 and throughout: I always appreciate when authors don't just state that there is an effect of X factor, but also state in what direction this effect is.

Thank you – we totally agree! It makes it much easier as a reader to understand the results if the magnitude and direction of the result are clear in the text.

Line 207: Not sure I quite understand “homogeneity in the level of familiarity amongst group-mates” – aren't both types of groups (familiar and unfamiliar) “homogenous” in their level of familiarity, in that they are all either very familiar with each other, or not familiar at all? The reference to homogeneity makes me think that some individuals in the group are more/less familiar with each other than others.

Thank you for pointing this out. You are absolutely right and we can understand why this language is misleading. We have edited this text to instead read “degree of familiarity amongst group-mates”. (Lines 236-237)

Methods: I'll just note here that I found their methods well described and easy to follow, especially their

statistical analyses (which I normally am very nit-picky about). Just one quick question – on Line 441 the authors state that they include school as a random effect in their models testing whether turning rate and distance covered of first responders differed across school types. But the response variable is of first responders, so shouldn't there only be one value per school

This is a great point, and something that we have now clarified in the manuscript. We included the random effect in the first-responder models related to average turning rate and distance covered to account for the tied first responders in many schools (as unlike with latency, first responders did not exhibit identical values for these traits). We have clarified this point in the “Statistics and Reproducibility” section of the methods (lines 485-487) and also added in further details about the frequency of ties when this phenomenon is first described in the results (lines 150-152), as well as in the legend for Table S1 in the supplementary information (where the number of responders by responder number are detailed for the familiar and unfamiliar treatments).

Reviewers' Comments:

Reviewer #1:

Remarks to the Author:

The revised manuscript has addressed my concerns. This is an interesting study that adds to the growing literature on the social cognitive abilities of fishes.

A minor point that I missed in my review the first time is the structure of the sentences in the abstract. While I appreciate the authors providing explanatory notes in parentheses, this is done at a high frequency in every sentence, sometimes more than once in a sentence. The author(s) may wish to consider including the explanatory note directly in their sentences within the given word space.

Signed,

Ajay S. Mathuru

Reviewer #2:

Remarks to the Author:

I was reviewer 2 on an earlier version of this manuscript and I'm happy to say the authors have fully addressed my concerns. I really like their revised introduction as its now clear why it might be expected to be a link between familiarity and fast-start responses (through changes to the reactivity of particular neurons). This really improves the paper, I think!

The only minor completely nit-picky thing is that I'm still not crazy about suggesting that fish shouldn't have social memory because they don't have 'advanced cognitive abilities' in the abstract. The authors themselves cite many studies that looked at exactly this - the effects of familiarity/social interactions on behavior in fish no less - so making this statement is not even in line with what the rest of the manuscript (let alone scientific evidence) says. It just seems like a bit of a strawman. The authors might consider removing this from the final version.

Reviewer #1

The revised manuscript has addressed my concerns. This is an interesting study that adds to the growing literature on the social cognitive abilities of fishes.

A minor point that I missed in my review the first time is the structure of the sentences in the abstract.

While I appreciate the authors providing explanatory notes in parentheses, this is done at a high frequency in every sentence, sometimes more than once in a sentence. The author(s) may wish to consider including the explanatory note directly in their sentences within the given word space.

Signed,

Ajay S. Mathuru

Thank you for this comment. We had used this sentence structure to save on our word count but agree on re-reading the abstract that the frequency of this practice can be awkward. To address this issue, we have removed the parentheses for several of these instances and instead incorporated the text into the sentence. Particularly, see changes on lines 24, 25, and 31, as well as the revision to the parenthetical

statement in lines 27-28.

Reviewer #2

I was reviewer 2 on an earlier version of this manuscript and I'm happy to say the authors have fully addressed my concerns. I really like their revised introduction as its now clear why it might be expected to be a link between familiarity and fast-start responses (through changes to the reactivity of particular neurons). This really improves the paper, I think!

The only minor completely nit-picky thing is that I'm still not crazy about suggesting that fish shouldn't have social memory because they don't have 'advanced cognitive abilities' in the abstract. The authors themselves cite many studies that looked at exactly this - the effects of familiarity/social interactions on behavior in fish no less - so making this statement is not even in line with what the rest of the manuscript (let alone scientific evidence) says. It just seems like a bit of a strawman. The authors might consider removing this from the final version.

We can understand your perspective on this sentence, so have instead revised this statement to say the following (lines 32-34):

“These findings demonstrate that the benefits of social recognition and memory may enhance individual fitness through greater survival of predator attacks.”